# Color-complexity enabled exhaustive color-dots identification and spatial patterns testing in images

**Shuting Liao**[1], **Li-Yu Liu**[2], **Ting-An Chen**[2], **Kuang-Yu Chen**[3], **Fushing Hsieh**[4]*

**1** Graduate Group in Biostatistics, University of California at Davis, Davis, CA, United States of America, **2** Department of Agronomy, National Taiwan University, Taipei, Taiwan, **3** GEOSAT Aerospace & Technology Inc., Tainan, Taiwan, **4** Department of Statistics, University of California at Davis, Davis, CA, United States of America

* fhsieh@ucdavis.edu

**Data Availability Statement:** All data files are available from the Dryad database (DOI: 10.25338/B8WG9S).

## Abstract

Our computational developments and analyses on experimental images are designed to evaluate the effectiveness of chemical spraying via unmanned aerial vehicle (UAV). Our evaluations are in accord with the two perspectives of color-complexity: color variety within a color system and color distributional geometry on an image. First, by working within RGB and HSV color systems, we develop a new color-identification algorithm relying on highly associative relations among three color-coordinates to lead us to exhaustively identify all targeted color-pixels. A color-dot is then identified as one isolated network of connected color-pixel. All identified color-dots vary in shapes and sizes within each image. Such a pixel-based computing algorithm is shown robustly and efficiently accommodating heterogeneity due to shaded regions and lighting conditions. Secondly, all color-dots with varying sizes are categorized into three categories. Since the number of small color-dot is rather large, we spatially divide the entire image into a 2D lattice of rectangular. As such, each rectangle becomes a collective of color-dots of various sizes and is classified with respect to its color-dots intensity. We progressively construct a series of minimum spanning trees (MST) as multiscale 2D distributional spatial geometries in a decreasing-intensity fashion. We extract the distributions of distances among connected rectangle-nodes in the observed MST and simulated MSTs generated under the spatial uniformness assumption. We devise a new algorithm for testing 2D spatial uniformness based on a Hierarchical clustering tree upon all involving MSTs. This new tree-based *p*-value evaluation has the capacity to become exact.

## Introduction

Spray technologies via unmanned aerial vehicle (UAV) for liquid chemicals of fertilizers, herbicides and pesticides are at the stage of intensive research and developments [1]. From economic and environmental perspectives, these technologies are deemed vital in Precision Agriculture [2]. Since its wide uses will not only save costs from many aspects, particularly on

**Funding:** The raw images were collected within a facility belonging to a commercial company: GEOSAT Aerospace & Technology Inc. This private company provided support in the form of salaries for author [K.-Y. C.], but did not have any additional role in the study design, data collection and analysis, decision to publish, or preparation of the manuscript. CEDAR Seed funding supports for F. H.

**Competing interests:** We declare that the author [K.-Y. C.] has a commercial affiliation with the private company GEOSAT Aerospace & Technology Inc. through employment. This study does not involve with any consultancy, patents, products in development, or marketed products of this company. This commercial affiliation does not alter our adherence to PLOS ONE policies on sharing data and materials.

human labor and illness, but also add capabilities of dynamic and optimal management. However, the success of such technologies heavily relies on effective evaluations of their performances in terms of efficiency and precision. There might be many ways of making such evaluations. One fundamental way is to evaluate their performances by testing whether the sprayed liquid droplets are relatively homogeneous in size and distributed in a spatially uniform fashion upon a target area.

Recently it has become very common that companies and research labs design their own experiments to facilitate such fundamental testing. One key step of this testing involves color image analysis consisting of two coupled computational tasks: exhaustive color-dots identification and spatial patterns extracting and testing. These two tasks are in accord with two natural perspectives of color complexity of a color image [3]. It is crucial to be able to exhaustively identify all targeted color dots of all sizes on a target area. Since each dot of sprayed chemical gives rise to two pieces of information: its amount of chemical and spatial location. Exhaustive search and extraction often are difficult to achieve computationally. Even though color identification is a major topic in computer science, those publicly available techniques, such as Contour on grayscale and other color segmentation techniques through computer package like OpenCV, are not optimal, nor practical choices for dealing with heterogeneous shading on color images. In order to better perceive and appreciate the color complexity and its induced computational challenges facing us in this study, some relevant information of the physical nature of color is essential as well as necessary.

Humans recognize different colors when visible lights are received by photoreceptors. Those colors human perceive are indeed grouped into categories, but due to other processes and not only photoreceptors' response. In the physical world, the region of the wavelength of the rainbow of visible lights is between 780nm to 380nm (in decreasing order at nanometer scale). That is, Red has the longest wavelength (around 700nm) and the smallest frequency (428 Terahertz (THz)), while Purple has the shortest wavelength (380nm) and the largest frequency (714THz). The Yellow is in between with wavelengths around 580nm and frequency around 517THz. Such linear ordering of wavelength and frequency of visible light is turned into a circular order of colors, called the color wheel, through human perception. Isaac Newton has studied this nonlinearity in the 17th century. Now we know that the three types of cone cells in our human eyes, which specifically respond to three visible lights: Red, Green and Blue, collectively generate all colors shown on a color wheel [4–6].

How the lights come from the outside world into our eyes or lens of cameras is the topic of the physical nature of color. Outside of the eyes and camera, the color should be described exactly through the spectral reflectance within the wavelength region from 380 nm to 780 nm. This spectral reflectance and diffuse reflection [7, 8], depend on the nature of the material and its surface properties, light source, viewing angles, observer (the properties of eyes or camera for imaging) and other surrounding objects. Light not only reflects from the surface of material, but also penetrates beneath the surface and then scatters. Blue has the highest scattering intensity than that Red and Green due to its short wavelength and high frequency. How visible lights are composed into colors inside eyes and cameras is the topic of human trichromacy. The RGB color space is resulted from technical convolution of three monochromatic spectral stimuli: $R(\lambda)$, $G(\lambda)$, $B(\lambda)$, curves [4–6]. This RGB color model is mutually transformable with the HSV model: Hue, Saturation and Value [9]. HSV and its variant models were designed and are popularly used by computer graphics researchers [10].

Besides RGB and HSV models, there are color systems being used with a focus on various color characteristics in different topics and industries. As for color-printing techniques, two color-systems are popularly used for different purposes. The CMYK (Cyan (close to blue), Magenta (close to red), Yellow and Black) system is primarily used in a printing factory, the

PMS (Pantone Matching System) system is designed to identify exact color needed, such as in a paint shop. This study is in the opposite direction. We try to avoid or at least limit involvement of human visualizations.

In contrast with RGB, which was defined and adopted in 1930, CIELAB color system: $L*a*b*$, was defined by the same International Commission of Illumination in 1976 [11]. Here $L*$is referred to as lightness: white-vs-black. $a*$ is indexed relative to green-vs-red, while $b*$ is indexed relative to blue-vs-yellow. Its original intention is to approximate human perception. To convert RGB or CMYK coordinates to or from $L*a*b*$ is not straightforward. We must know the reference Illuminant of RGB and CMYK beforehand. This requirement is not practical for images taken in the field study. It is well known that CIELAB lacks perceptual uniformity, particularly around the blue hues. This character of CIELAB surely raises a special concern, especially for using it as a color system in this study. Therefore we mainly focus on RGB and HSV in this paper.

With respect to a chosen color model or system, a color image is indeed a large data matrix. Theoretically, any data set has a Kolmogorov complexity in Information Theory. This complexity is referred to as the shortest length of a computer program that can regenerate the data as one whole [12]. Though this is not a computable concept, it is highly relevant to the majority of data analysis involving finding its pattern-based information content, such as in this study.

Unlike the concept of Kolmogorov's complexity, the color complexity is simply considered through human's data visualization, not computers' data generation. This is a concept not yet being well established in Color Theory. A colored image is a huge dataset because of its many thousands or millions of pixels, and each pixel is coded with 3D RGB and HSV coordinates. The color complexity of an image can be perceived and computed from two perspectives: color variety and color distribution [3].

In this paper, both perspectives are being studied across five experimental images. The color variety is referred to how many "distinct" colors are present within an image, while color distribution is referred to 2D spatial distributions of distinct colors on the 2D Euclidean plane of the image. We discuss these two perspectives computationally in this study. Even though, each of the five images seemingly has only two major colors by experimental design, the color variety within the RGB space surely would be much more than 2. There are many underlying factors that could affect the large and fine scales coloring of an image. We just list three key factors here as follows.

First, the mechanism of light's reflection upon an unsmooth and uneven colored paper media involves with complicated physical nature of color, as been briefly depicted with some basic facts above. Secondly, colored dots are of many sizes, and indeed could contain heterogeneous color-pigment intensities. Thirdly, swiftly varying lighting conditions could go through the lens of cameras when the images were taken. Due to these reasons and beyond, we expect that one image would give rise to a RGB point-cloud occupying many 3D coordinates within the RGB space. We can intuitively define an image's color variety as the relative size of its RGB point-cloud on the finest scale with respect to the size $256^3$ of 3D RGB space $[0, 255]^3$.

In this study, we analyze five images collected by the researchers from GEOSTA & Technology Inc., which conducted the experiments of spraying liquids via drone. The experimental setting realistically mimicked the mechanism undertaken by an unmanned aerial vehicle (UAV) when spraying liquid chemicals on the fields. Multiple yellow Teejet water-sensitive papers (part# 20301-1N) are laid on flat ground when an UAV machine was employed at each trail. Such yellow paper shows stained blue-purplish colors wherever being exposed to aqueous spray droplets. The five testing papers that are analyzed in this paper are collected from five different spraying trails with distinct mechanical parameters that control spraying mechanisms. Water is used to do the test and there was no pesticide involved. More information on

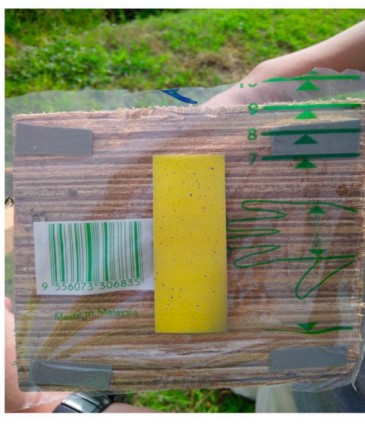
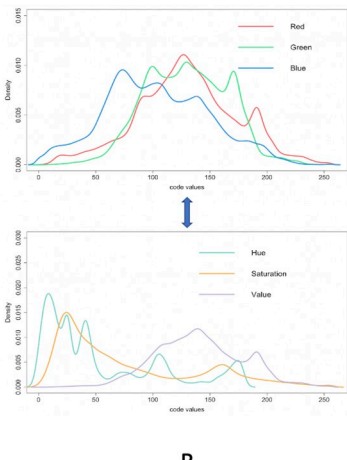

|  A  |  B  |

**Fig 1. Original image and RGB density curves.** A: Original image with purple dots in various sizes and shapes located on the yellow test paper; B: RGB model and HSV model.

the design of the UAV machine and nozzle's spray angle-vs-coverage relational information is contained in [13].

After spraying, each test paper was then individually placed on a specific background-setting. Its camera image was taken. The five background-settings are slightly different from each other. Therefore, the five camera images of the five testing papers were taken under different lighting conditions. For instance, see Fig 1A for one experimental image. Then, each of the five images is individually converted into one RGB and one HSV data sets, see Fig 1B for both color models. Within these five pairs of data sets, the joint characteristics of spatial geometries and size distributions of blue-purplish colored dots are used to evaluate the effectiveness of the five experimental mechanisms.

Thus, the goals of this study are to be able to computationally extract precise spatial and distributional information of all blue-purplish dots and to prescribe their interacting relations as joint characteristics. To achieve these goals, blue-purplish color identification is the first computational technique we would develop in this paper. This technique needs the capability of capturing color-dots of all sizes with varying shapes. Based on RGB and HSV color models, we expect that each of the five images would have a rather small color complexity. It is because that the two major colors would make three RGB coordinates as three features of 3D data points highly associated. That is, all RGB point-clouds are to have a two-hill shape occupying small volumes in the RGB space.

Various algorithms and techniques such as thresholding techniques [14] and clustering, have been developed for image or color segmentation [15]. Most of them can be implemented through OpenCV. Applying color thresholding typically requires human involvement in color identification by looking for spatial spots in an image that match a fixed range of colors. Such a range of colors is either pre-specified or being taken out by humans from an image. Such a priori kind of knowledge makes the most direct and clear distinction between existing popular color-Identification packages, such as OpenCV, and goals of color Identification in this study. Since this range of color within an image is one specific information to be computed and learned from the image itself. That is exactly one of the challenges facing us in this study. Since the heterogeneity in targeted color created by varying shades and tones makes specifying such a precise range of colors extremely difficult.

Specifically, shades and tones affect colors locally, while lighting conditions would impact colors globally. Such lighting conditions are nearly uncontrollable for any camera image taken outdoor under the Sun. Such global and local factors make color distributions vary drastically from one image to another. This varying nature does not fit well with many color-identification techniques based on computational methodologies, such as K-means [16] and variants of thresholding techniques [17], that require prior knowledge of object and background distributions. Further, irregular shapes are another characteristic of the color dot in our testing papers. Such irregularity in shape and varying in sizes together with a large number of dots make the majority of segmentation techniques [15] nearly useless. The widely spreading tiny color dots would act like noise that seriously compromise the effectiveness of such segmentation techniques.

The second computational technique is to characterize spatial geometries with respect to varying dot-sizes as the targeted joint characteristics. As for the perspective of color dot distribution, we only focus on the targeted purple color's distribution because the Yellow color is the background color. Since the spraying mechanism is a mechanical one. So, we expect the potentials that the purple color's distribution might be far away from uniform, particularly for dots of large size with high intensity of purple. Due to the largeness of the number of purple color-dots, we focus on spatial uniformness in the sense of density of purple dots, not their spatial coordinates. In order to bring out the sense of density, we divide the test paper into a feasible number (400) of squares, and categorize their densities in terms of their distributions of categorical sizes of purple dots contained in them. For one cumulative density category, from the largest to smallest, we propose to build a minimum spanning tree (MST) to connect squares. A MST is a rather flexible spatial structure. Its characteristic distribution of the distance between immediate neighboring-squares would be the basis for testing spatial uniformness.

By simulating MST under spatial uniformness assumption upon all squares, we compare the observed MST with all simulated MSTs. To facilitate such a comparison, we extract one MST-based distribution of the distance of neighboring squares from each MST. Further, we transform each distance distribution into a histogram with common data-driven bin-boundaries, and then collecting all vectors of proportions into a matrix. We build a Hierarchical Clustering (HC) tree among these distribution-IDs. Then, we develop a new algorithm to calculate $p$-value based on the binary structure of HC tree-geometry. We then repeatedly perform the same testing on uniformness by including less dense squares in a cumulative manner. This $p$-value computation is somehow novel in the sense that it is calculated based on a series of odds-ratios along a descending tree-path leading to the observed MST-tree-leaf. Such HC-tree-based spatial-uniformness testing and evaluating the reliability of the testing results are performed with respect to each of the multiple cumulative density scales. We believe that such a multiscale spatial 2D uniformness testing seemingly offers a relatively new perspective of spatial data analysis.

In comparison, our goals and computational developments for evaluating the effectiveness of an UAV's spraying mechanisms in this study are rather unique comparing with existing color-identification techniques. For instance, in medical image analysis, the focus is placed on identifying color-dots under a rather limited and well-controlled range of lighting conditions. Therefore, the issues of shade and tone are not as serious as in the open field settings. Further, color images are highly sensitive to environmental and operational conditions, such as lighting and shadowing under weather and operations. It is realistic, practical and even necessary to extract designated color pixels with respect to data-driven RGB ranges, not fixed ones as used in popular color identification approaches, such as the aforementioned color thresholding and

others. Since the effects of shade and tone vary from one color image to another color image, we need robust computational efforts to accommodate such varying effects.

## Method

The original dataset is in the format of an image, with a dimension of $4608 \times 3456$. We use R programming to read and load the image, transforming it into a 3-dimensional array. Each dimension corresponds to a $4608 \times 3456$ matrix with one color channel as its entries. We further reconstruct the data as a large matrix with x-, y-axes, and three color-channels as columns. We make use of the array-like object to reproduce an image and illustrate our results, and a matrix for computation.

Each image gives rise to two RGB and HSV data formats, as shown in Fig 1B. They are mutual convertible. We remove most of the background and only focus on the area containing the yellow test paper, which contains around $10^6$ pixels. The image of the test paper can be reconstructed as any one of RGB or HSV $10^6 \times 3$ matrices. That is, from perspectives of pixel-wise 3D intensities of RGB and HSV, a color image data here is in the form of structured data. That is, each pixel specific color data point in an image like the one in Fig 1A) are referred to two 3D color measurements: RGB ($[0, 255) \times [0, 255) \times [0, 255)$) and HSV ($[0, 255) \times [0, 255) \times [0, 255)$).

Before our developments in this section, we mention the fact that the five images are likely to subject to mixing lighting conditions. Such conditions can cause uniform shading effects across the entire image. Such uniform effects can be dealt with by taking off via various imaging processing techniques. Nonetheless, we do not perform such a data pre-processing step. Since we see that the shading effects are seen and considered in this paper are primarily local and heterogeneous. They are mainly caused by unevenness of the test paper, or some surrounding objects, such as human hands holding the test paper, that indeed block lights from shining onto small parts of the test paper when it was posted for its images to be taken. Such characteristics of locality and heterogeneity on an image are different from the uniform ones.

Our machine learning-based approach is proposed to identify localities affected by the shading effects and to figure out the varying degrees of the effects in a divide-and-conquer fashion. Further, such local and heterogeneous characteristics of shading effects vary from one image to another image. Therefore, we need at least a learning region being anchored for each image to train our computing methodologies on typical areas, and then modify our learned methodologies one way or the other to adapt to shading effects upon affected localities. We believe that the local and heterogeneous nature of shading effects, as being present across all five images studied here, are not uncommon in real-world applications.

We propose computational algorithms to resolve this color identification issue. One physical fact of color theory plays an important role: the three dimensions of RGB or HSV are highly associated. This fact interestingly and very important points to that, among the $256^3$ unit cubes (of size $1 \times 1 \times 1$), a color image's millions of pixels typically only collectively occupy a very small number of cubes. That is, a natural color image usually has a very small "color-complexity".

The color-complexity of the test paper in Fig 2A is 0.002. In contrast, the color-complexity of the whole image in Fig 1A is calculated as $\frac{393014}{256^3} \approx 0.023$. That is, the color-complexity of this test paper is only one-tenth of the original image. Therefore, we indeed deal with only several thousand, not a million, of distinct colors. This is the underlying reason why our computing cost is low and our color identification is effective.

Upon one test paper (yellow colored-strip), under either RGB or HSV color models, our first development is aimed at exhaustively identifying all "purple" dots of all sizes. Here color

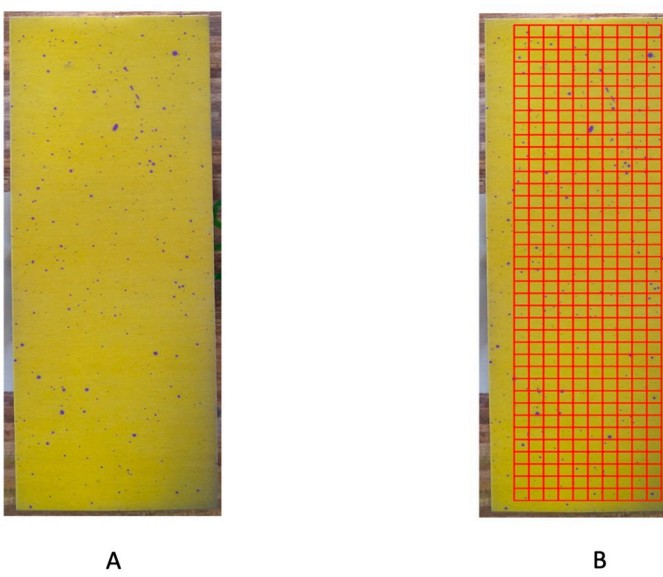

**Fig 2. A: Reduced image; B: Focal area.**

"purple" is meant to be an unspecified 3D region within the 3D discrete space $[0,255]^3$ that reveals purple color through our visual system. After nearly exhaustively extracting all purple pixels, we build purple dots as a connected network based on a common choice of the neighborhood on the 2D lattice. Then, we measure each dot's size, and classify them into several size categories.

After collecting almost all purple dots and sorting out their multi-scales size-categories, two key difficulties facing us here are: the largeness of the number of color-dots and geometric representations of color-dots. These two difficult aspects correspond to two kinds of uniformness: dot-size and spatial. Upon dot-size uniformness, we aim to figure out whether the spraying machine's mechanical design is proper or not. We particularly pay attention to the behavior of the right tail of the dot-size distribution (large and very large ones). While, upon the spatial uniformness, we need a practical unit that can embrace effectively the concept of spatial density of dot-locations. We also need a simple enough geometric representation to embed all involving units, so that structural information of spatial distribution can be extracted. Such computational endeavors for multi-scale spatial pattern extractions and testing spatial uniformness upon all size-scales are computationally developed in this study.

In this paper, we develop computing algorithms to resolve such coupled computational tasks. A block of diagram clarifying our proposed method process is displayed in Fig 3. We apply our algorithms to five experimental images under rather distinct lighting conditions. We exclusively use one image for illustrating our computational developments (Fig 1A), followed by the results of the rest of the 4 images.

In order to exhaustively identify all pixels of a human-designated color, under the RGB system, we apply the Hierarchical Clustering (HC) algorithm as an unsupervised machine learning approach in a divide-and-conquer fashion due to the largeness of pixel numbers. The tree-leaves of the designated color-branches are recovered with their spatial coordinates. The effectiveness of this color-identification can be inspected via zoom-in and zoom-out visual validations based on local as well as global geometries. Multiscale inspections conducted to make sure our algorithms' validity and efficiency are necessary due to the huge number of pixels.

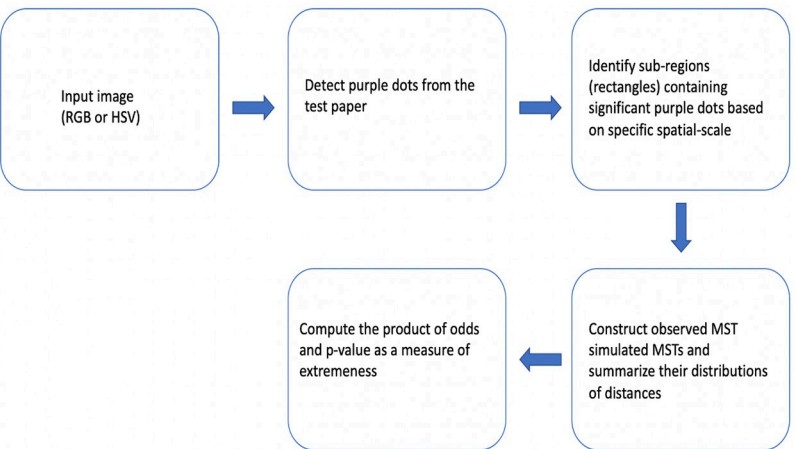

**Fig 3. A block of diagram of the process of our method.**

Further, since the collection of identified irregular shapes and size color dots is also large in number, we divide the whole spatial region into rectangles of the same size and extract their categorical features of color-dot-density. Then two brand-new computational approaches are devised for our data analysis. First, we propose to construct one minimum spanning tree (MST) upon one cumulative scale of color-dot-density to capture spatial characteristics. Secondly, we develop an odds-ratio-based algorithm to compute a *p*-value from a clustering-tree, which is built upon a set of summarizing vectors extract from the observed MST and many simulated MSTs under the uniformness hypothesis.

### Identification of purple pixels

As shown in Figs 1A and 2A, it is clear that this test paper contains two main color families, yellow and purple, among the one million ($10^6$) pixels. It is also evident that it contains areas of heterogeneous intensities of shades across the entire test paper. The presence of such data complications is rather common in the majority of real-world color images, It becomes part of the nature of data from Precision Agriculture. Since images might be taken under drastically distinct lighting conditions: with or without sun lights across different parts of days. Further, it is well known that the human's visual system via the brain and eye is subject to color illusions. Such an illusion makes us identify the same object with different colors under shadows as well as different backgrounds. Thus, any heterogeneously shaded image, in general, poses various challenges on color identification. One of the challenges is: How to do color identification in a data-driven fashion? In other words, it is a necessary capability of identifying color in any image from the perspective of a computer, not a human.

To make computing feasible via computer, we need to have an idea of how many distinct colors are indeed contained in the test paper. This is the concept referred to as "color-complexity". Given the discrete nature of color data, it is crucial to ask: how many unit cube of $1 \times 1 \times 1$ among the $256^3$ "color-unit-cubes" are indeed occupied by the one million color-pixels in the test paper? The answer is 28126. So the color-complexity is only $\frac{28126}{256^3} \approx 0.002$. If we enlarge the scale of the unit cube to a scale of $10 \times 10 \times 10$ cube, we checked and found that all potential colors contained within such a cube are still rather "uniform" to our raw eyesight. And, with respect to all pixels in the test paper, there are 880 among $26 \times 26 \times 26$ of such cubes being occupied. The color-complex of this test paper on this larger scale is $\frac{880}{26^3} \approx 0.05$. Hence, we decide to begin our machine learning computations upon this scale first and go back to the

unit scale afterward. It is worth emphasizing that such low color-complexity is made possible by very high non-linear associations among R&G&B and H&S&V. This is the underlying foundation to build data-driven algorithms for color identification.

Then we build a geometry among these 880 uniform color cubes. This geometry is intended to serve as a platform for our color identification. We choose this geometry to a tree for computational simplicity and practical applicability. We construct hierarchical clustering(HC) trees as follows. We use the center of mass (3D average) of pixels contained in such a cube as the cube's representative. Upon this collective of 880 representatives, the HC algorithm can work efficiently.

For completing our protocol of color identification, we take a step to tentatively avoid shady areas and background noises. Even though only involving a minority of pixels, their inclusion could yield non-negligible errors. To this end, we choose a rectangle area within the test paper as our "focal area", as shown in Fig 2B. This focal area is divided into 39 rows. Each row contains $2.5 \times 10^3$ pixels (Fig 2B), and is further divided into 10 squares.

Our color identification begins with the following row-by-row operation. For each one row's $2.5 \times 10^3$ pixels, we identify which color-cube it belongs to and then find its color-cube representative. The resultant set of distinct color-cube representatives has a size smaller than 880, surely is much smaller than $2.5 \times 10^3$. Upon this row-specific set of color-cube representatives, we apply the HC algorithm. For each row-specific HC-tree, via its bifurcation, we collect the representatives within the smaller branch as being designated as "purple" ones, while those in the larger branch as being "yellow". We further use ROC curve analysis for validation checking to avoid misclassification due to uncontrolled environmental and lighting conditions. This validation check is performed upon each square within each row.

The ensemble of color-identification on the focal area via the RGB data file is shown in Fig 4 together with results of the square-by-square validation check. There are three squares that have obvious misclassification. We "clean" these three squares by assigning all pixels in these three squares into the yellow group.

## Comparing with existing approaches

As aforementioned, one test paper likely includes regions under varying shading conditions due to the photographing condition and experimental setup when the image was taken. Heterogeneous shaded images are vividly seen as well in the four images shown and analyzed later. Such existential shading will complicate choices of grayscale, and consequently, reduce the efficiency of the Contour approach significantly. We apply these aforementioned methods and algorithms to one part of our image, as comparisons to our proposed method.

The segmentation results are shown in Fig 5 that indicates one example of such local impacts of the shading effect. It is observed that the shading effect exists along the right bound, especially at the bottom right. All three methods are sensitive to noise and shading, and thus fail to correctly segment the purple objects and yellow background. The poor identification by K-Means and the Mean thresholding method are affected due to the shading. As for the color thresholding method, we set the RGB code as (247,191,252) as the upper bound (light purple) while (75,0,130) as the lower bound (dark purple). Since so many dots of varying sizes look purple, it is not possible to specify a 3D color region to cover all purple dots.

We, therefore, conclude that the color thresholding technique via OpenCV is neither feasible nor practical for the exhaustive search purpose in our case. Hence, due to practical issues caused by shading and lighting effects, such classical applications are less effective, They neglect key color structures contained in the image. Thus, an effective strategy as proposed in the above subsection becomes necessary.

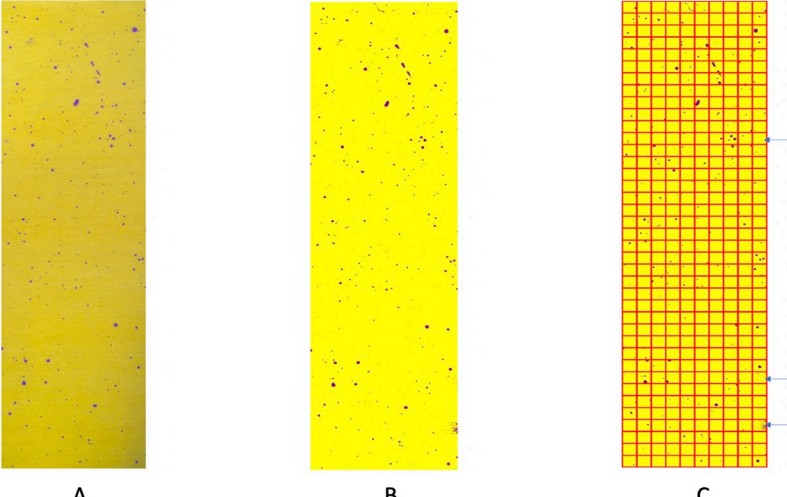

**Fig 4. Color-identification on the focal area.** A: original focal area; B: predicted focal area by RGB file; C: validation check indicating three squares need "cleaning".

## Recovering the whole yellow test paper

Given that pixels outside of the focal area have a higher potential for being subject to shade or other noises, their color identification needs extra effort. We propose the following remedy based on our experiences derived from our explorations and experiments. Upon RGB data format, we need to employ $1 \times 1 \times 1$ small RGB color-cubes, denoted as the scale of "$n = 1$". That is, we need to drastically sharpen the color-uniformness within each color-cube. So we have to pay more computing cost to achieve the goal of color identification with RGB data, even though we still enjoy the reduction of color complexity because only 0.2% of $1 \times 1 \times 1$ RGB unit color-cubes are occupied.

Consequently, we collect the centers of all color-cubes, which have ever been occupied by an identified purple pixel in the focal area. And likewise collect the centers of all color-cubes,

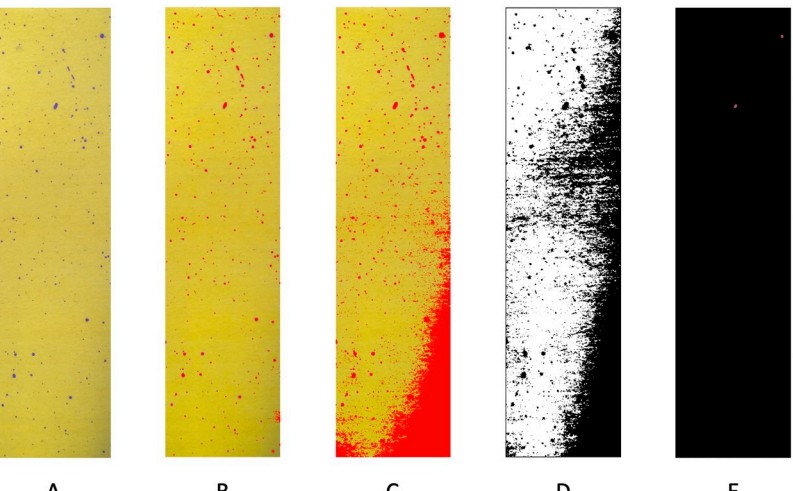

**Fig 5. Color segmentation using different algorithms and techniques.** A: part of the original image; B: our proposed method; C: K-Means; D: Mean thresholding; E: Color thresholding.

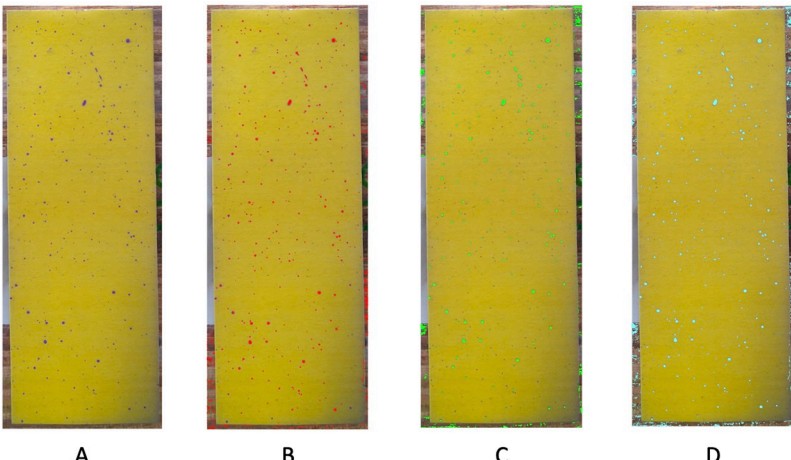

**Fig 6. Performance by using different files.** A: original reduced image; B: recovering (in red) by RGB file ($n = 1$); C: recovering (in green) by HSV file ($n = 10$); D: recovering (in light blue) by combining RGB file ($n = 1$) and HSV file ($n = 10$).

which have ever been occupied by an identified yellow pixel in the focal area. Among the pool of these two collections of centers of color-cubes, we compute the closest neighbor to each pixel outside of the focal area and then declare the color-identification accordingly. In this way, we are able to capture the majority of purple pixels and avoid misclassification as much as possible.

As for HSV data format, we still employ the scale $n = 10$, i.e. $10 \times 10 \times 10$ HSV color-cubes. We obtain the recovering by both the RGB file and HSV file separately. It turns out that the RGB file helps identify more pixels in smaller purple dots, while HSV file helps identify more pixels near the bottom and top where the RGB file fails. The two results suggest a better recovering scheme as simply combining these two results together. All results are shown in Fig 6.

## Testing uniformness via sizes

As it is intuitively known that a spraying device typically mixes air with liquid, and then pushes the mixture out. The mixing of air and liquid is determined by a set of tuning parameters. Mechanically speaking, different sets of tuning parameters surely give rise to distinct degrees of inhomogeneous mixing. Consequently, the droplets out of the device are likely heterogeneous in size. So, some tuning parameters are better than others. One merit of exhaustive identification of targeted color-dots contained in an image is to check the validity of a parameter-setup of the spraying device. For this merit, there are two natural measure sizes of a droplet, which is an identified connected purple-pixel. The first measure is to count the number of connecting pixels. The second one is the radius of the smallest circle containing all connecting pixels. Accordingly, the best set of tuning parameters should ideally produce the Poisson distribution with respect to the counting measure, and an Exponential with respect to the continuous measure.

We consider the target collection of color-dots identified via the approach of combining the RGB and HSV data, see Fig 6D. We first compute the MLEs of intensity parameters, $\lambda_P$, and $\lambda_E$, under the Poisson and Exponential distribution assumptions, respectively, based on the two data sets derived from the target collection of purple-dots within the test paper.

Based on the pixel-count data set, the Poisson distribution specified by MLE of $\lambda_P$ is computed and superimposed onto the histogram constructed based on pixel-counts from the target

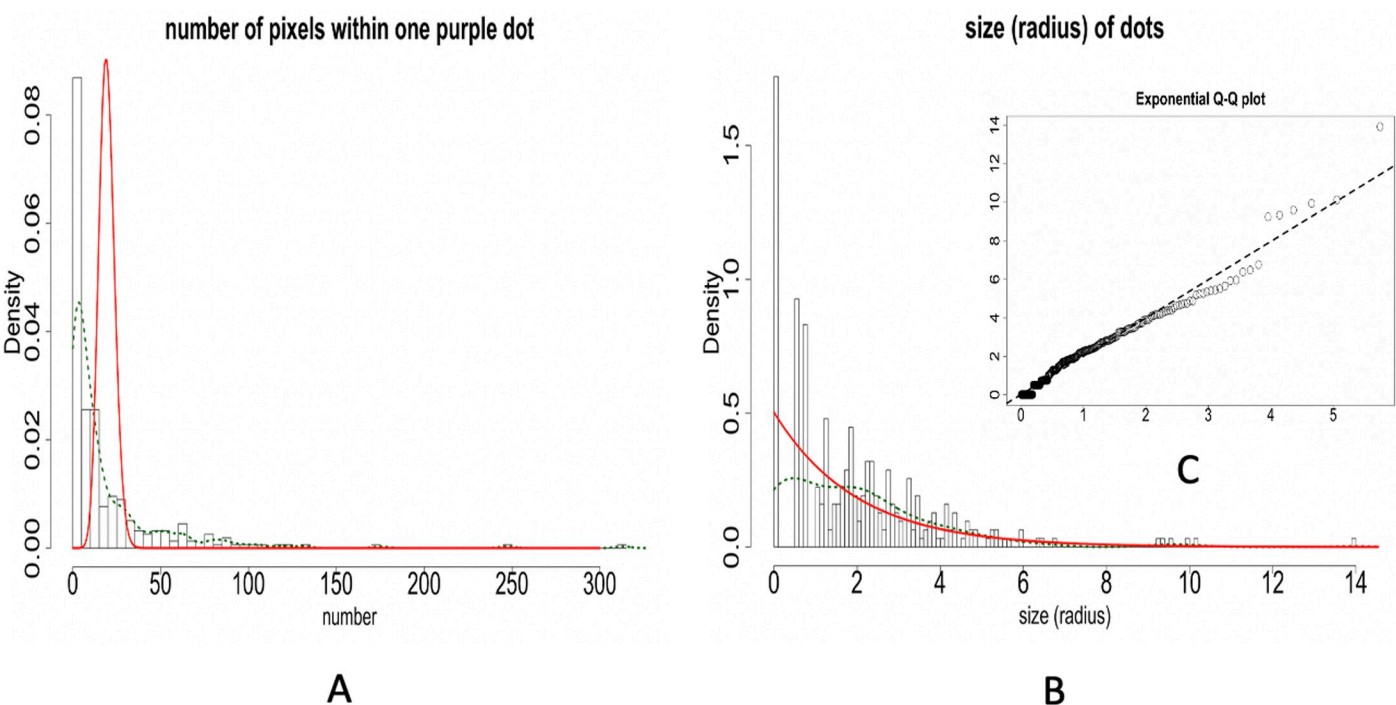

**Fig 7. Distributions of pixel-counts and dot-sizes.** A: The histogram of pixel-counts of identified purple-dots superimposed with Poisson distribution specified MLE of $\lambda_P$(red curve) and kernel density estimates (green dash curve); B: The histogram of dot-sizes of identified purple-dots superimposed with Exponential distribution specified MLE of $\lambda_E$ (red curve) and kernel density estimates (green dash curve); C: Empirical Q-Q plot (Circle curve) vs Exponential Q-Q plot specified by MLE of $\lambda_E$(dash line).

collection of purple dots, as shown in Fig 7A. It is evident that many identified purple-dots have large pixel counts that can not be accounted for by Poisson distribution. We can draw a similar conclusion based on the dot-size distribution with superimposed Exponential distribution specified by MLE of $\lambda_E$ shown in Fig 7B.

While the Q-Q plot Exponential distribution specified by MLE of $\lambda_E$ is compared with empirical Q-Q plot of continuous purple dots sizes, as shown in Fig 7C. We see evident departures from this Q-Q plot comparison. Further, we run the Kolmogorov-Smirnov test, which suggests the observed dot-size is not following an Exponential distribution (with $p$-value $<0.05$).

## Testing spatial uniformness via rectangle neighborhood

In this section, we construct our major algorithmic developments for testing against the 2D spatial uniformness. We adopt the concept of the 2D neighborhood into 2D spatial characteristics. The reasons behind this are that the number of identified purple-dots is too big, and their sizes are rather heterogeneous. This neighborhood concept directly links to the idea of spatial density, which is a proper expression for addressing spatial uniformness here.

Given that we specifically divide the entire target area into 400 small rectangles, one rectangle is taken as one 2D neighborhood. On this collective of rectangles, we pretend as if they are uniformly colored with an intensity (density) of purple depending on all purple-dots contained in it. In this fashion, we consider the 2D spatial uniformness among 2D-entities of 400 rectangles. In addition, the radius of the smallest circle containing all connecting pixels is regarded as the size of a purple dot and thus all identified purple-dots will be classified into

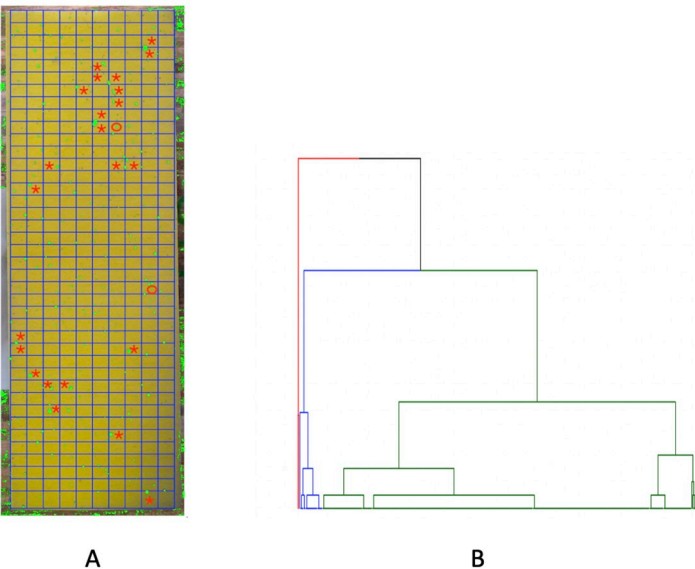

**Fig 8. Significant rectangles are selected by using HC algorithm.** A: ⋆ in red: rectangles with ≥ 1 big dot are identified (in blue branch in B); *o* in red: rectangles with ≥ 2 medium dots (in red branch in B); B: HC tree of 400 rectangles with 3 indicated clusters.

three categories of sizes: small, medium and large, according to their sizes. So intensity of purple dots in a rectangle would be also categorized, as given below. We apply the Hierarchical Clustering (HC) algorithm to guide this categorization. The categorizing protocol is devised as follows.

We count the numbers of small, medium and large purple-dots contained in a rectangle as the 3 features for this rectangle. That is, each rectangle of many pixels as an unstructured data format is characterized by a 3-dim vector of counts. Via this characterization, we transform a rectangle into a structured data format. We employ a distance measure that is a weighted version of Euclidean distance in $R^3$. To reflect larger dot-size giving rise to higher purple-color intensity, this weighting scheme is specified with respect to the 3 averaged sizes: small, medium and large, of purple-dots. With this weighted distance measure, we build a $400 \times 400$ distance matrix. A HC-tree is computed and reported in Fig 8B.

Upon this HC-tree, we can see two small branches (red and blue colored) constituting a clear pattern: their member rectangles either contain at least one large or two medium dots. Locations of these rectangles are shown in Fig 8A. This data-driven pattern leads us to explore the intensity spectrum via a Hierarchical Clustering (HC) tree on these 400 rectangles.

More or less based on this HC tree, we further qualitatively determine four categories of density within a rectangle as follows: A rectangle contains

**R1:** [Highly-dense] one or more large dots, or two or more medium dots;

**R2:** [Dense] one medium dot;

**R3:** [Sparse] 2 or more small dots;

**R4:** [Extremely-Sparse] only one small dot or empty.

We found that there are 25 rectangles belonging to the [R1] (Highly-dense) category, which are located on the blue and red branches of the HC tree in Fig 8B. The spatial geometries of these four categories of rectangles can be seen in Fig 9 in a cumulative manner.

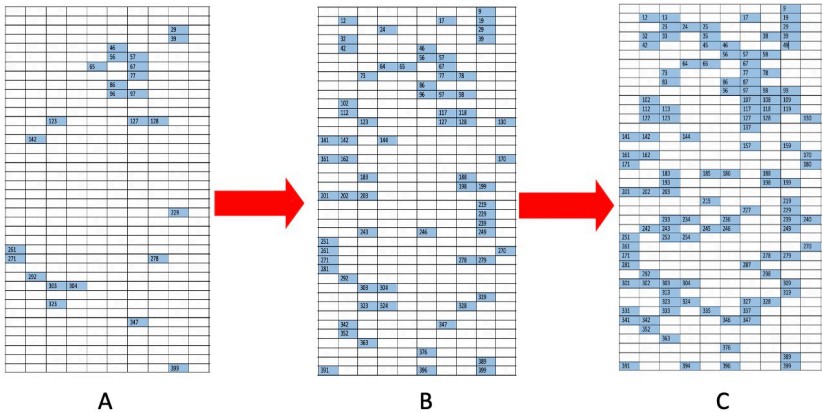

**Fig 9. The flow of rectangles' 2D-distribution, from large dots to medium and small dots; some specific spatial patterns are present.** A: [R1] rectangles; B: [R1-R2] rectangles; C: [R1-R3] rectangles and the blank are [R4] rectangles.

## Via Minimum Spanning Tree (MST)

Upon these 25 highly-dense rectangles, we construct a Minimum Spanning Tree (MST), denoted as $M^{obs}$, as shown in Fig 10B. The intuitive idea underlying MST is that its tree geometry, which spans a subregion by having one tree-leaf linking to one of its close neighbors, will reflect possibly heterogeneous degrees of spatial concentration among the 25 rectangle members. One way of expressing such heterogeneity in the spatial concentration of a MST is to look through the empirical distribution (or histogram) of distances among all connected immediate-tree-neighbors. Such an empirical distribution (or histogram) is an informative summarizing exhibition for the degree of heterogeneous concentration pertaining to a MST. We particularly lookout for the extremely high concentrations, which will lead a MST's empirical distribution of distance, or its histogram, to reveal a single-mode located at a small distance value.

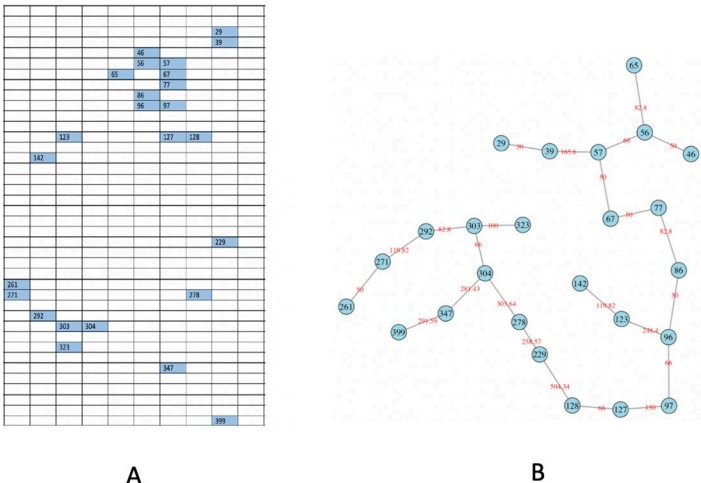

**Fig 10. Distributions of [R1] (Highly-dense) rectangles and construct MST.** A: interested 2D distribution of rectangles with $\geq 2$ medium dots; B: using the corresponding Minimum Spanning Tree (MST) to capture the spatial pattern.

With aforementioned focal characteristics in mind, to test whether $M^{obs}$ is coherent with the 2D spatial uniformness hypothesis, we compare $M^{obs}$'s empirical distribution (or histogram) with $B(= 500)$ randomly generated MSTs' empirical distributions (or histograms). 25 numbers are sampled randomly from a collection of digits {1, 2, . . .,400} with equal probability. We repeat this simulation scheme for $B(= 500)$ times with independence. We accordingly generate corresponding $B$ MSTs, denoted as $\{M_b\}_{b=1}^B$. So we have $B$ simulated empirical distributions (or histograms) under the spatial uniformness hypotheses. To compare $M^{obs}$ with $\{M_b\}_{b=1}^B$ via their empirical distributions (or histograms), we propose two approaches: the Receiver Operation Characteristic (ROC) curve analysis and unsupervised machine learning approach.

### ROC curve analysis

The ROC Curve compares a pair of distributions, say $F(.)$ and $G(.)$ via $1 - G(F^{-1}(1 - w))$ with $F^{-1}(w)$ the quantile function of $F(.)$. So we compute $B$ ROC curves for the $B$ pairs empirical distributions pertaining to $B$ pairs of $(M^{obs}, M_b)$ with $b = 1, . . .,B$. So $B$ pieces of area-under-curve (AUC) are calculated. The histogram of $B$ AUC values are shown in Fig 11A. The marked 0.5 value of AUC, which corresponds to the case of $F(.) = G(.)$, is seen being far away from this histogram. This fact strongly indicates that the $M^{obs}$'s empirical distribution (or histogram) is stochastically smaller than the empirical distributions (or histograms) of $\{M_b\}_{b=1}^B$ in a persistent manner. Nevertheless, one fundamental drawback rests on the fact that one-dimensional statistics is unlikely to reveal structural differences between two distributions because of their high dimensionality. That is why unsupervised machine learning approaches are needed.

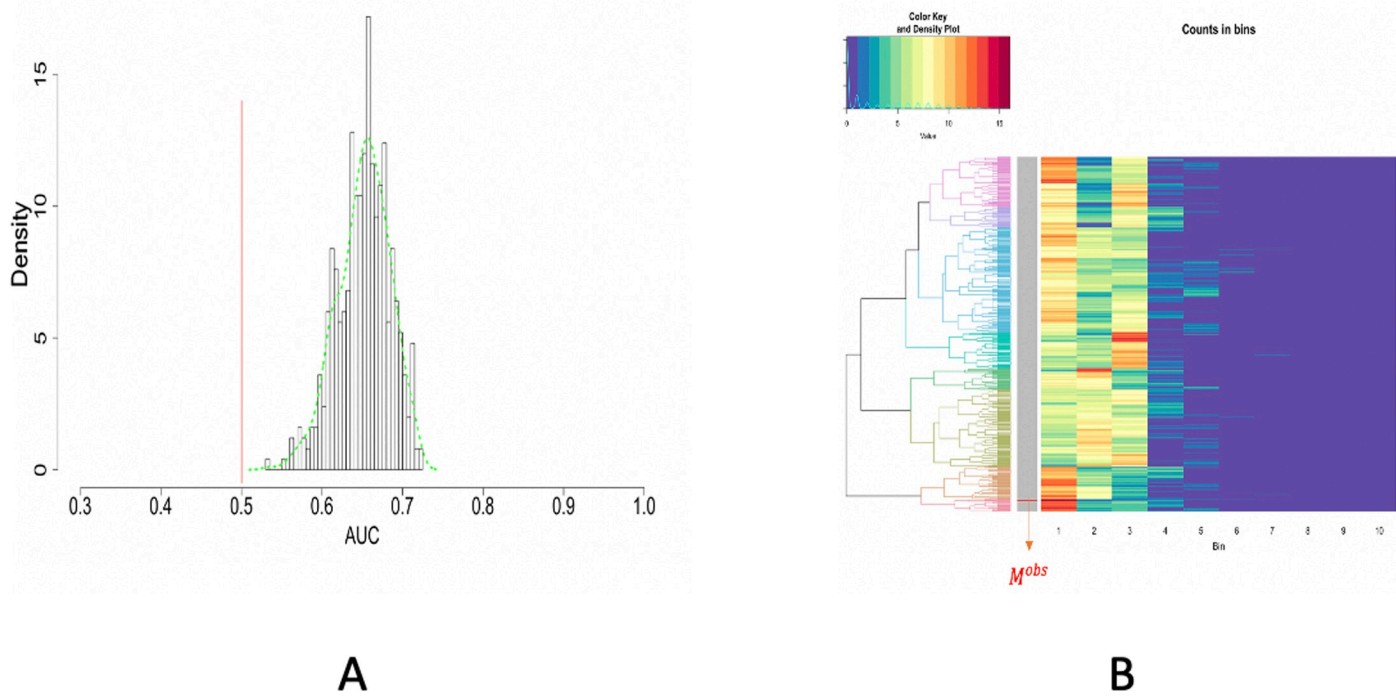

**A**

**B**

**Fig 11. ROC curve and unsupervised machine learning approach on testing the spatial uniformness of 25 highly-dense rectangles.** A: ROC curve anlysis results; B: HC algorithm with heatmap, product of odds ($PO = 0.023$) and $p$-value $p(M^{obs}) = 0.002$.

## Unsupervised machine learning approach

We want to literally compare these $B+1$ empirical distributions derived from $M^{obs}$ with $\{M_b\}_{b=1}^B$. To facilitate such a direct comparison, we pool together all distance values from these $B+1$ empirical distributions and then build a histogram with 10-bins. With such data-driven bin boundaries, we transform each empirical distribution into a 10-dim vector of counts. These $B+1$ 10-dim vectors are arranged along the row-axis a $(B+1) \times 10$ matrix.

Due to the equal total counts on all rows, we simply adopt Euclidean distance and then calculate a $(B+1) \times (B+1)$, with which we apply the Hierarchical Clustering (HC) algorithm to build a HC tree, denoted as $\mathcal{T}$, and superimpose it onto the row-axis of $(B+1) \times 10$ matrix, as shown in Fig 11B. The tree-leaf of $M^{obs}$ is marked onto the HC tree upon this rearranged matrix, which is called a heatmap.

The resultant heatmap explicitly shows why $M^{obs}$ is found among an extreme subgroup of the ensemble $\{M_b\}_{b=1}^B$. The visible pattern is that the 25 members of $M^{obs}$ have dominantly many extremely small distances among immediate neighbors. This pattern indeed indicates high degrees of concentration among 25 members of $M^{obs}$. This is a strong piece of evidence against the spatial uniformness assumption. How strong it is? Next, we develop an algorithm to do such an evaluation.

The HC tree $\mathcal{T}$ is binary. Therefore each of $B+1$ tree-leave can be located by a binary tree-descending tracing process. If we adopt a coding scheme to encode the left-branching with a code-0 and right-branching with a code-1 at each internal node of $\mathcal{T}$. Then each tree-leaf is encoded by a binary code sequence. Denote the binary code sequence for $M^{obs}$ as $< d_1^o, d_2^o, ... d_{K_o}^o >$ and a code sequence for $M_b$ as $< d_1^b, d_2^b, ... d_{K_b}^b >$ with $b = 1, .., B$. The coding lengths $K_o$ and $\{K_b\}_{b=1}^B$ vary from one tree-leaf to another tree-leaf.

Further, with the binary code sequence as the descending path of bifurcating for locating $M^{obs}$, we denote the left and right branches at $k$-th bifurcation as $L_{d_k^o}$ and $R_{d_k^o}$ with $k = 1, ..., K_o$. The sizes of the two branches are denoted as $|L_{d_k^o}|$ and $|R_{d_k^o}|$. Then the size of the branch containing $M^{obs}$ at $k$-th bifurcation is calculated as

$$|L_{d_k^o}|^{(1-d_k^o)} |R_{d_k^o}|^{(d_k^o)}. \tag{1}$$

Then there is an odds of correctly guessing which branch contains $M^{obs}$ is calculated as:

$$PO[d_k^o | M^{obs}] = \frac{|L_{d_k^o}|^{(1-d_k^o)} |R_{d_k^o}|^{(d_k^o)}}{|L_{d_k^o}|^{(d_k^o)} |R_{d_k^o}|^{(1-d_k^o)}}. \tag{2}$$

We then compute the overall odds of guessing correctly on which branch $M^{obs}$ belongs along the entire coding sequence $< d_1^o, d_2^o, ... d_{K_o}^o >$ as

$$PO(M^{obs}) = \prod_{k=1}^{K_o} PO[d_k^o | M^{obs}]. \tag{3}$$

An example is illustrated in Fig 12.

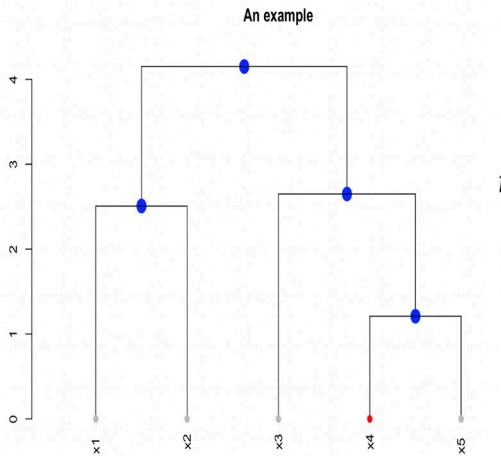

**Fig 12. An example of the product of odds and *p*-value.**

Likewise we compute an ensemble of odds $\{PO(M_b)\}_{b=1}^{B}$. Then we compute the *p*-value of observing an odds like $PO(M^{obs})$ as the proportion of $PO(M_b)$ being less than $PO(M^{obs})$:

$$p(M^{obs}) = \frac{\sum_{b=1}^{B} 1_{[PO(M_b) < PO(M^{obs})]}}{B}. \qquad (4)$$

Upon the HC tree shown in Fig 11B, we have $PO(M^{obs}) = 0.023$ and *p*-value is $p(M^{obs}) = 0.002$. Hence, it turns out that $M^{obs}$ is significantly extreme in the HC tree. Based on the visible patterns observed in the heatmap, we can conclude that the 25 [R1] (Highly-dense) rectangles are not uniformly distributed.

## Enlarge the spatial scale

In the previous section, we have built a MST for the Highly-dense rectangles' 2D-distribution, as the 3rd of the flow chart shown in Fig 9C. We further work on the 2D distribution of the 4th one including those Dense rectangles, as shown in Fig 13A. By combining the [R1] and [R2] scales of rectangles, we expect to see the distribution of purple-dots with an expanded

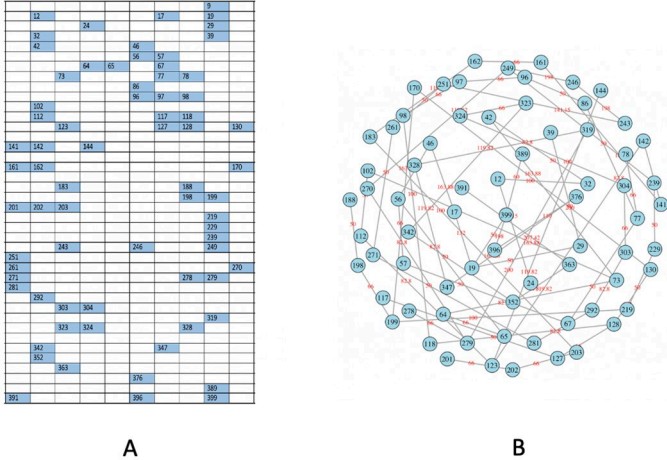

A                                                    B

**Fig 13. Enlarge the spatial scale.** A: focusing on [R1-R2] rectangles; B: the corresponding MST.

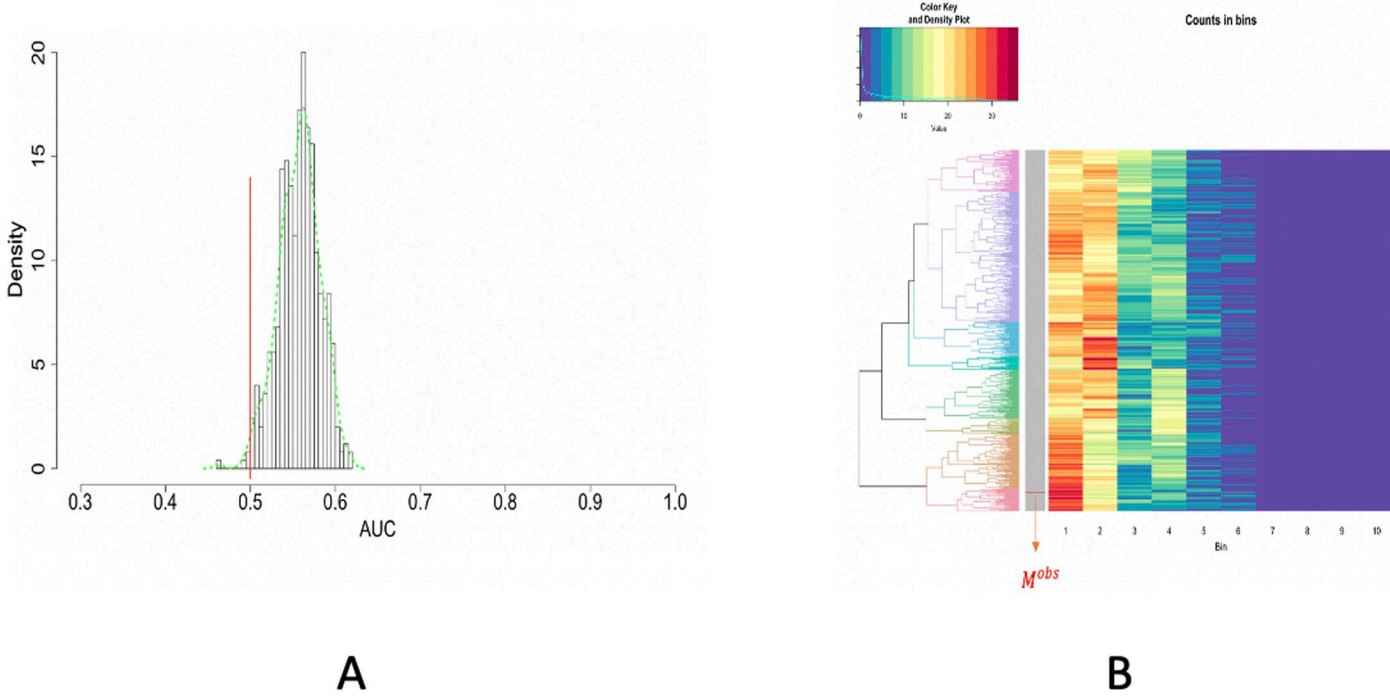

**Fig 14. ROC curve and unsupervised machine learning approach ontesting the spatial uniformness of [R1-R2] rectangles.** A: ROC curve analysis results; B: HC algorithm with hearmap, product of odds ($PO = 1.108$) and $p$-value = 0.056.

perspective. Its MST is computed and reported in Fig 13B. Likewise, we perform the two versions of testing on spatial uniformness and report the results in Fig 14, respectively. Though having less significance in terms of $p$-values, the results via ROC curve analysis on the left panel and Machine Learning method on the right panel all still indicate that the rectangles' 2D distribution is not fully in accord with spatial uniformness.

Nonetheless, this trend of getting less and less significant against spatial uniformness is expected when we further expand by including rectangles of [R3] Sparse scale. This trend tells us that the spraying mechanism needs further fine-tuning in order to achieve spatial uniformness. Especially, large purple nodes in [R1] Highly-dense scale should be significantly reduced.

## Simulation study

In addition to the application to the real data, we conduct experimental simulations to validate our proposed method for 2D spatial uniformness testing via rectangle (or sub-region) neighborhood given known spatial scale densities. Two situations are considered in Fig 15. We divide the area into 400 rectangles. In simulation 1 (Fig 15A), all rectangles at spatial scale [R1-R4] are randomly uniformly distributed. In simulation 2 (Fig 15B), we select rectangles at [R1] around the center and bottom. Rectangles connected to those at [R1] are chosen as [R2] with a probability of 0.4. Similarly, rectangles connected to [R2] are selected as [R3] with a probability of 0.5. We sample [R4] rectangles from those connected to [R3] with a probability of 0.6. For each case, we apply our algorithm to three scales: Highly-dense ([R1]), Highly-dense with Dense ([R1-R2]), and Highly-dense with Dense and Sparse ([R1-R3]). The results are reported in Table 1 and consistent with our simulated data. The $p$-values and $PO$ of three spatial scales in simulation 1 are all relatively large, indicating the uniformness of purple

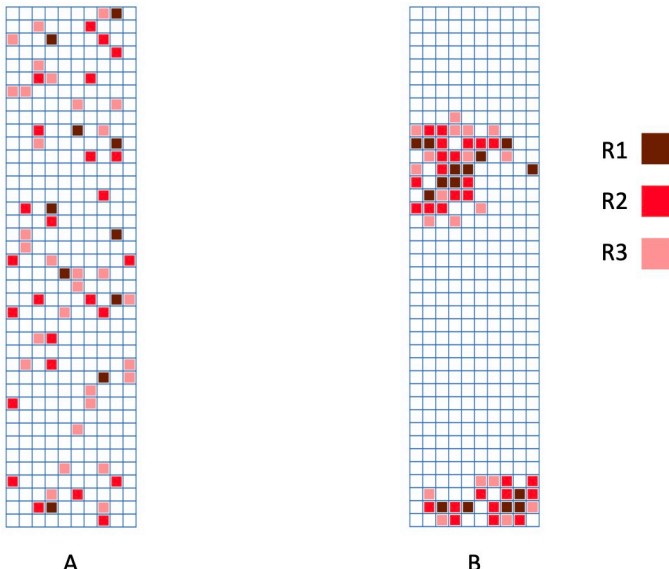

**Fig 15. Two simulations.** A: simulation 1—purple rectangles are uniformly distributed; B: simulation 2—purple rectangles are clustered in two sub-regions.

distribution. In simulation 2, a non-uniformness distribution is suggested according to small *p*-values.

## Analysis of 4 other images

We apply computational algorithms developed and illustrated through image no.1 to the rest of the four images. Heterogeneous shading conditions can be evidently seen across these four images. Overall our exhaustive color identifications are satisfactory and the testings of spatial uniformness are indeed much more effective than the one based on ROC analysis.

### Image no.2

Image no.2 consists of two "pages" of test papers, as shown in Fig 16A. The upper part of these two coupled test papers is under shading. Consequently, the color-dot identification based only on RGB has missed quite a few small purple dots, as shown in Fig 16B. Many of these small dots were also not picked up via HSV data format based on $1 \times 1 \times 1$ fine-scale cubes.

Based on the observation as well as *p*-values indicated, the purple dots are not distributed uniformly and improvement in spraying via drones is needed. We separately report results of spatial uniformness on the two test papers by focusing only dense squares as shown in Fig 16C. Two separate results are reported: Fig 17 for the Left and Fig 18 for the Right, respectively. Based on both figures, we see that there exists a small discrepancy in *p*-values between the result based on ROC in Fig 17A against results based on HC-tree and heatmap in Fig 17B, and

**Table 1. Computed results for two simulations.** *p*-values with *PO* in parentheses are reported.

|              | [R1]             | [R1-R2]          | [R1-R3]           |
|--------------|------------------|------------------|-------------------|
| Simulation 1 | 0.558 (20397.38) | 0.418 (64.48407) | 0.934 (222504.8)  |
| Simulation 2 | 0.008 (0.03275434) | <0.0001 (0.002) | <0.0001 (0.002)   |

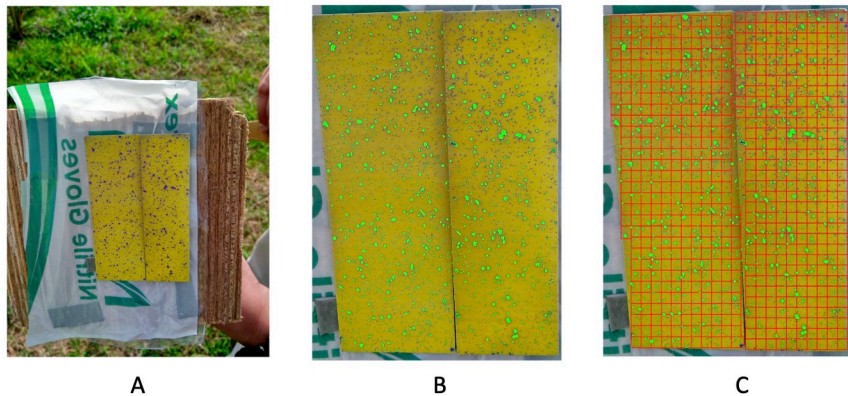

**Fig 16. Image no.2: A: Two test papers in the original data; B: Recovering by RGB file; C: Dividing the paper into rectangles for 2D spatial uniformness testing.**

result based on ROC in Fig 18A against results based on HC-tree and heatmap in Fig 18B. However, such small discrepancies don't seem to cause incoherent conclusions.

## Image no.3

A conclusion of uniform distribution of purple dots can be confirmed, according to the $p$-value. The test paper in image no.3 seems curved a bit, as shown in Fig 19A. This curved shape likely created shads around the upper left and lower right corners of the test paper. The coupled results from RGB and HSV seem to achieve a big degree of exhaustive identification

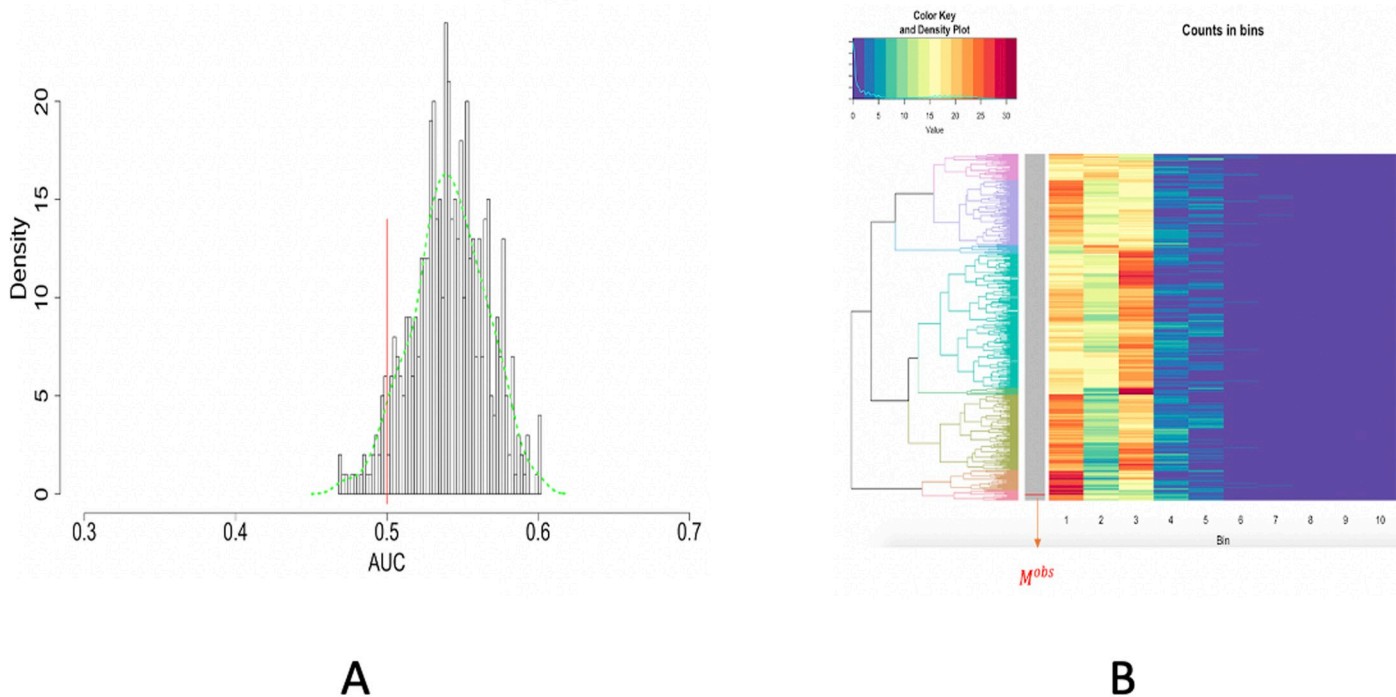

**Fig 17. Image no.2: Spatial uniformness testing on the LEFT test paper: Studying [R1] rectangles; A: ROC curve anlysis results; B: HC algorithm with heatmap, product of odds ($PO$ = 0.016) and $p$-value $p(M^{obs})$ = 0.**

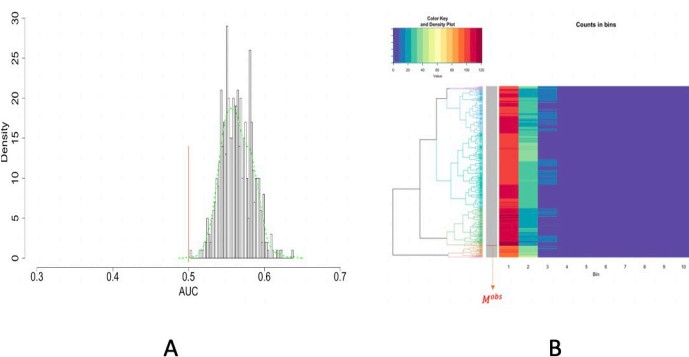

**Fig 18. Image no.2: Spatial uniformness testing on the RIGHT test paper: Studying [R1] rectangles; A: ROC curve anlysis results; B: HC algorithm with heatmap, product of odds ($PO$ = 0.047) and $p$-value $p(M^{obs})$ = 0.004.**

except dots locating the two corners, as shown in Fig 19B. For the spatial uniformness test, the result based on ROC analysis, as shown in Fig 20A, seems to point to a direction being slightly different from the one based on HC-tree and heatmap, as shown in Fig 20B. Given the observed row being well mixed with simulated one within a big branch, we are more confident on the $p$-value result based on HC-tree and heatmap Fig 20B. According to these results, we can conclude that the drone is satisfying.

### Image no.4

Image no.4 has obvious shades at the upper and lower boundaries of the test paper, as shown in Fig 21A. We report the color-dot identification result based on HSV cubes of $n$ = 10 scale, as shown in Fig 21B. For spatial uniformness testing, a big gap is seen between the result based on ROC analysis and the one based on HC-tree and Heatmap, as shown in Fig 22A and 22B, respectively. Again the latter result seems more reliable. Consequently, we are satisfied with even spread of purple dots based on a relatively large $p$-value.

### Image no.5

Image no.5 looks like it is being twisted a bit, particularly at the lower right corner, and the shades are visible all over, as shown in Fig 23A. The final color-dot identification is based on

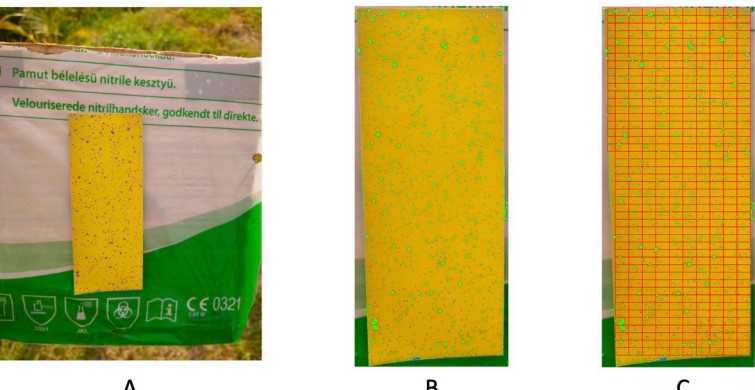

**Fig 19. Image no.3: A one test paper in the original data; B recovering by 6D [RGB+HSV] file ($n$ = 10); (c) dividing the paper into 479 rectangles for 2D spatial uniformness testing.**

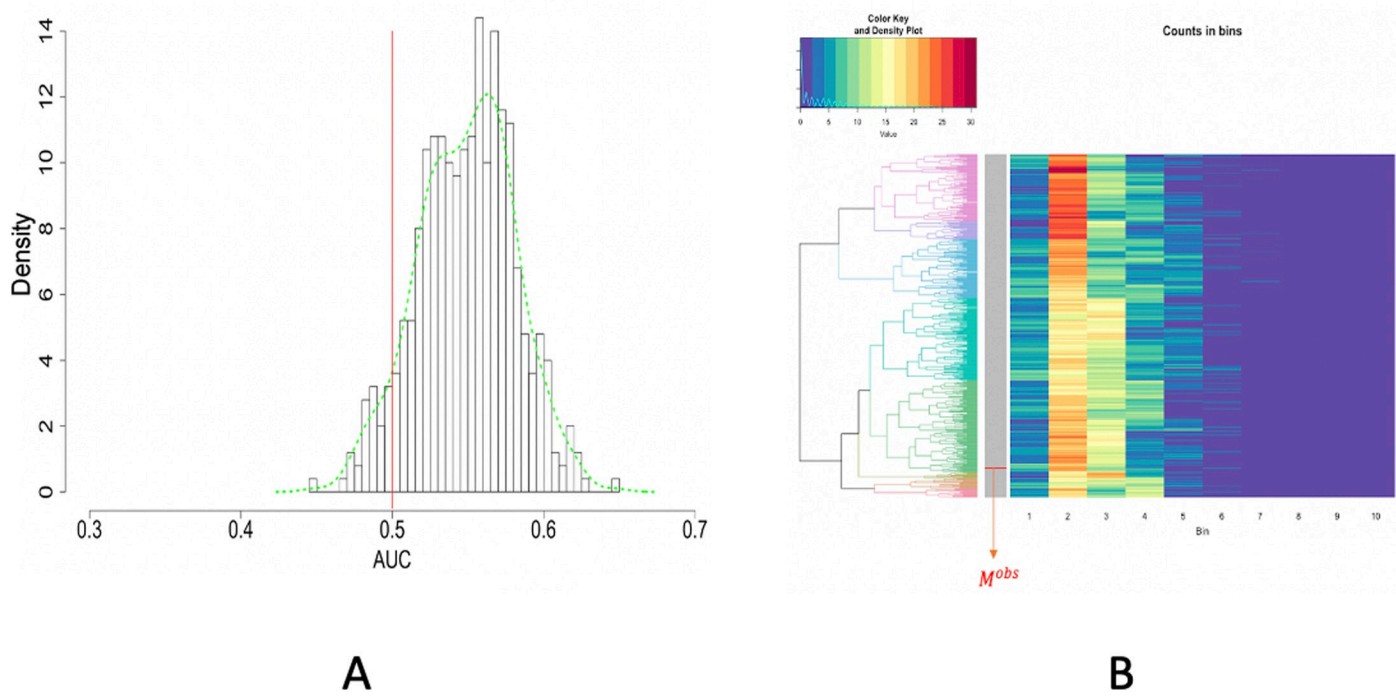

A                                                    B

**Fig 20. Image no.3: Spatial uniformness testing based on [R1] rectangles; A: ROC curve anlysis results; B: HC algorithm with heatmap, product of odds** ($PO$ = 1164.056) **and** $p$-**value** $p(M^{obs})$ = 0.686.

the coupled results of RGB and HSV, as shown in Fig 23B. The discrepancy between the two results of spatial uniformness testing is especially wide. But, based on the small size of the branch containing the observed row vector, we have more confidence in the one based on HC-tree and heatmap, as shown in Fig 24B, over the ROC one, as shown in Fig 24A. The distribution of purple dots is marginally uniform because of the relatively small $p$-value, which indicates a fair job done by the drone.

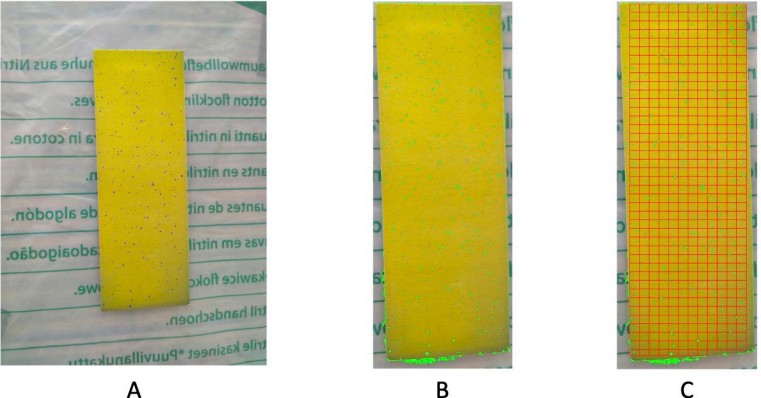

A                          B                          C

**Fig 21. Image no.4: A: One test paper in the original data; B: Recovering by HSV file (**$n$ **= 10); C: Dividing the paper into 488 rectangles for 2D spatial uniformness testing.**

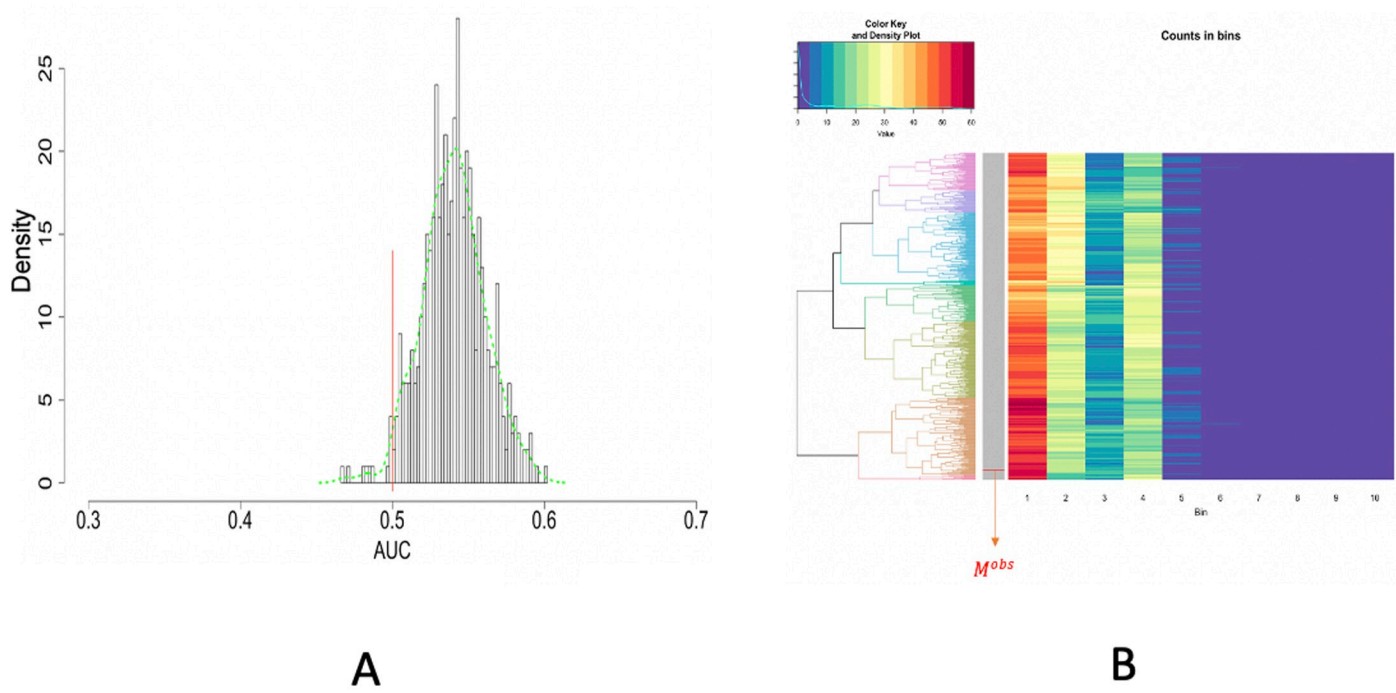

A                                                                                                    B

**Fig 22. Image no.4: Spatial uniformness testing based on [R1] rectangles; A: ROC curve anlysis results; B: HC algorithm with heatmap, product of odds ($PO = 193.157$) and $p$-value $p(M^{obs}) = 0.642$.**

## Conclusions

The color-identifications and testing 2D spatial uniformness via MST for the five images are rather satisfactory. Basically, these results collectively strongly indicate that our data-driven computational approach for color-identifications is rather effective, and testing methodology for 2D spatial uniformness is novel and practical.

The underlying reason for the effectiveness of our color-identification approach is the low color-complexity. This interesting fact is that this simple concept is not well known in the literature. In fact, our current color research has shown us that low color-complexity is seen in

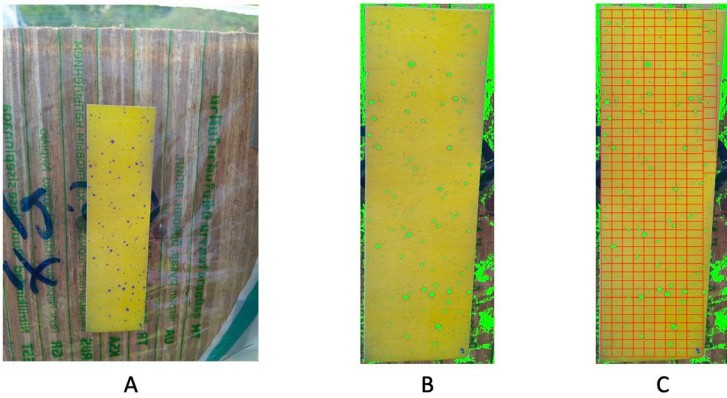

A                                    B                                    C

**Fig 23. Image no.5: A: One test paper in the original data; B: Recovering by 6D [RGB+HSV] file ($n = 10$); C: Dividing the paper into 384 rectangles for 2D spatial uniformness testing.**

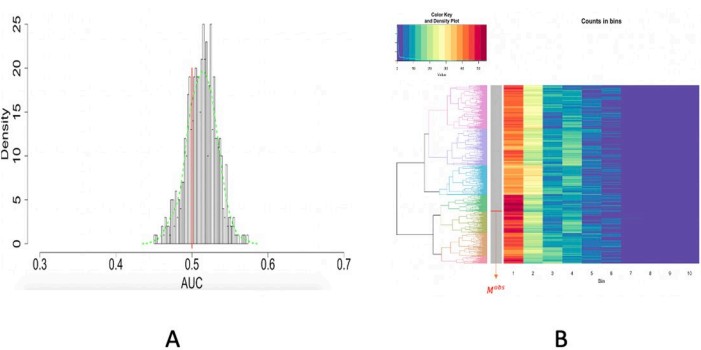

**Fig 24. Image no.5: Spatial uniformness testing based on [R1] rectangles; A: ROC curve anlysis results; B: HC algorithm with heatmap, product of odds ($PO$ = 0.334) and $p$-value $p(M^{obs})$ = 0.056.**

natural images as well as in images of famous paintings. That is, our data-driven color-identification is applicable in a wide range of color images.

Admittedly, employing stereo cameras with more single imaging angles is able to provide a better observation, even help eliminate the shading effect. Such Data can reveal the truly 3D-homogeneity of 3D-spatial distributions. Analysis of such 3D distribution data can resolve many realistic and essential issues facing precision agriculture, that is unlikely could be solved by combinations of RGB and HSV data of 2D images. However, provided with the limited 2D image-format in this paper, we try our best to extract as much information as we can from observed data sets. Given that RGB and HSV are two distinct aspects of the same images, they provide slightly distinct pattern information. Therefore, it is natural to combine both aspects of pattern information in hope of improving on results relying on solo representation.

The MST structure and its distance distribution are new and essential summarizing pattern information of spatial data. They are shown to be good approaches to illustrating and characterizing useful data structures in Data Science. The novelty of evaluating $p$-value via products of odds-ratios based on a tree structure, which is complex, can critically expand the applications of unsupervised machine learning methodologies to wider ranges of scientific fields, including the medical one.

Our way of dealing with shading in images is not sophisticated. We adopt the fact that RGB and HSV data formats could be differentially affected by shading. So, we propose to combine results from both data formats under distinct scales. From the results reported in Section 5, we see that it works with different degrees of success. It might be possible to develop systematic approaches to remove, or at least lessen the shading effects. This is one of our undertaking research directions right now.

## Author Contributions

**Data curation:** Li-Yu Liu, Ting-An Chen, Kuang-Yu Chen.

**Formal analysis:** Shuting Liao, Fushing Hsieh.

**Methodology:** Shuting Liao, Fushing Hsieh.

**Supervision:** Fushing Hsieh.

**Visualization:** Shuting Liao, Li-Yu Liu.

**Writing – original draft:** Shuting Liao, Fushing Hsieh.

**Writing – review & editing:** Shuting Liao, Fushing Hsieh.

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
