## [Decision Letter · Decision Letter 0]

11 Mar 2021

PONE-D-20-32597

Color-complexity enabled exhaustive color-dots identification and spatial patterns testing in images

PLOS ONE

Dear Dr. Fushing,

Thank you for submitting your manuscript to PLOS ONE. After careful consideration, we feel that it has merit but does not fully meet PLOS ONE’s publication criteria as it currently stands. Therefore, we invite you to submit a revised version of the manuscript that addresses the points raised during the review process.

I found this manuscript well written and interesting. As you will infer from below that there was disagreement among reviewers regarding enthusiasm for this work. Reviewer 1 and Reviewer 2 had made certain observations and recommended your work with major decision. However, Reviewer 3 recommended minor revision.

After thorough consideration of comments from all reviewers, I felt that your study has merit but identified points that need to be addressed. Therefore, my decision is “major revision”.

We look forward to receiving your revised manuscript.

Kind regards,

Gulistan Raja

Academic Editor

PLOS ONE

Journal Requirements:

2)  Thank you for stating the following in the Financial Disclosure section:

[The author(s) received no specific funding for this work.].   

We note that one or more of the authors are employed by a commercial company: GEOSAT Aerospace & Technology Inc.

i. Please provide an amended Funding Statement declaring this commercial affiliation, as well as a statement regarding the Role of Funders in your study. If the funding organization did not play a role in the study design, data collection and analysis, decision to publish, or preparation of the manuscript and only provided financial support in the form of authors' salaries and/or research materials, please review your statements relating to the author contributions, and ensure you have specifically and accurately indicated the role(s) that these authors had in your study. You can update author roles in the Author Contributions section of the online submission form.

ii. Please also provide an updated Competing Interests Statement declaring this commercial affiliation along with any other relevant declarations relating to employment, consultancy, patents, products in development, or marketed products, etc.  

3)  We note that you have indicated that data from this study are available upon request. PLOS only allows data to be available upon request if there are legal or ethical restrictions on sharing data publicly. For more information on unacceptable data access restrictions, please see http://journals.plos.org/plosone/s/data-availability#loc-unacceptable-data-access-restrictions.

Reviewers' comments:

Reviewer's Responses to Questions

**Comments to the Author**

1. Is the manuscript technically sound, and do the data support the conclusions?

Reviewer #1: No

Reviewer #2: Yes

Reviewer #3: Yes

2. Has the statistical analysis been performed appropriately and rigorously? 

Reviewer #1: No

Reviewer #2: No

Reviewer #3: Yes

3. Have the authors made all data underlying the findings in their manuscript fully available?

Reviewer #1: Yes

Reviewer #2: Yes

Reviewer #3: No

4. Is the manuscript presented in an intelligible fashion and written in standard English?

Reviewer #1: Yes

Reviewer #2: Yes

Reviewer #3: Yes

5. Review Comments to the Author

Reviewer #1: Please provide more context. For non-experts in Precision Agriculture, are these papers being laid in a field where spraying is to take place, and purple dye is mixed with the pesticides, and the papers are people examined for dye density? Can you give some references to help?

More references need to be given in general. For example, you mention "popular color identification approaches" and "OpenCV" but could you please be more specific about which techniques you're referring to and why they can't accommodate effects of shade and tone"? There are certainly robust techniques in medical imaging for example using color science, morphology, and image processing for segmentation. There are many techniques printing that you could reference for identifying dots on paper. Why not show an example of a simpler technique on your data for comparison, to motivate the improvements you are claiming with your technique? Your conclusion that "this simple concept [of color-identification approach for low color-complexity] is not well known in the literature" needs substantiation.

I think it would help the reader considerably to see a block diagram of your technique. I found it difficult to follow your explanation of the algorithm without some kind of "map." There are many algorithms mentioned and it was difficult for me to understand the progression. I admit I am not well-versed in the computer vision techniques you reference, which is again why more references would be helpful for the reader.

I can say that your technique seems to give good results but there are so many figures of purple dots on yellow paper. Is it possible to condense these to a final few results? I honestly got lost with all the discussion of 72, 62, 55, 142 rectangles. Again less of these, and one clarifying diagram of your process would go a long way.

Using a more sound color space seems appropriate. You mention that it won't be discussed and give a reference to a chapter in an online class, but without really explaining WHY you don't transform to CIELAB or another color space. Please discuss how "RGB and HSV data formats could be differentially affected by shading" as they are a linear transformation from each other. The color discussion in the introduction was confusing to me as a color scientist. A bit more visual information / diagrams to explain your points would be appreciation.

Finally, I did not see a measure of "goodness" other than visual inspection but perhaps that is the Q-Q plot? If so, please explain as this diagram in more detail. Would it be possible to create a simulated target to test your algorithm where you know for sure the purple density?

Reviewer #2: The ideas of the author are quite interesting and practical. However, the authors should read more about color science so that the issued algorithms are more appropriate with the physical nature of color. The shading effects and other possible noises can be solved correctly. The literatures and also the-state-of-the-art should be analyzed more carefully, so that the statement and conclusions are not subjective and in a hurry.

Reviewer #3: - Line 21: I wouldn’t refer to OpenCV as a technique, but rather a tool. Please refer to the exact technique or rewrite the sentence. Same stands for the rest of the paper where you refer to Open CV.

- Line 21: Citations for techniques are missing. In addition, Introduction should also include techniques that are not "publicly available".

- Line 28: Angles instead of angels.

- Some information from Introduction (mostly from line 48, parts where you explain your approach in details) should be in Method section. In this way you would avoid repeating the facts.

- Line 74: citation is missing for this statement. As well as for other parts of manuscript where exact measures (for example, for "colour-complexity") and very specific statements are used.

- Line 254: It was not mentioned before that dots were classified into three categories. Please state the criteria for classification.

- Line 305: I don’t quite understand why you suggest using Mann-Whitney statistics to compare distribution. As stated in the paper, the main assumption may not be fulfilled, and you also noted the lack of complexity. I suggest to leave this part out.

Pages 11 and 12: Please add some general comments regarding the uniformity of those images (whether it would be regarded as satisfactory or not).

6. PLOS authors have the option to publish the peer review history of their article (what does this mean?). If published, this will include your full peer review and any attached files.

Reviewer #1: No

Reviewer #2: No

Reviewer #3: No

---

## [Author Response · Author response to Decision Letter 0]

6 Apr 2021

Response letters ('Response to Reviewers.pdf' and 'Response to the Editor.pdf') have been uploaded with all the other files.

---

## [Decision Letter · Decision Letter 1]

23 Apr 2021

Color-complexity enabled exhaustive color-dots identification and spatial patterns testing in images

PONE-D-20-32597R1

Dear Dr. Fushing,

We’re pleased to inform you that your manuscript has been judged scientifically suitable for publication and will be formally accepted for publication once it meets all outstanding technical requirements.

Kind regards,

Gulistan Raja

Academic Editor

PLOS ONE

Additional Editor Comments (optional):

Reviewers' comments:

Reviewer's Responses to Questions

**Comments to the Author**

1. If the authors have adequately addressed your comments raised in a previous round of review and you feel that this manuscript is now acceptable for publication, you may indicate that here to bypass the “Comments to the Author” section, enter your conflict of interest statement in the “Confidential to Editor” section, and submit your "Accept" recommendation.

Reviewer #2: All comments have been addressed

Reviewer #3: All comments have been addressed

2. Is the manuscript technically sound, and do the data support the conclusions?

Reviewer #2: Yes

Reviewer #3: Yes

3. Has the statistical analysis been performed appropriately and rigorously? 

Reviewer #2: Yes

Reviewer #3: Yes

4. Have the authors made all data underlying the findings in their manuscript fully available?

Reviewer #2: Yes

Reviewer #3: Yes

5. Is the manuscript presented in an intelligible fashion and written in standard English?

Reviewer #2: Yes

Reviewer #3: Yes

6. Review Comments to the Author

Reviewer #2: Thank the author! They have understood and improved all of my comments. In fact, the mentioned issues can be the nich cooperation between their research and my study for a very good level of the color identification.

Reviewer #3: All the previously given comments have been addressed. Some omissions I noticed in this version are:

Sentence starting from Line 33 – Here categorical perception is linked directly to the photoreceptors, which is not quite correct (for example Himba tribe members do not have the category for blue). It would be more appropriate to state that humans recognize different colors when visible lights are received by photoreceptors. Those colors we perceive are indeed grouped into categories, but due to other processes and not only photoreceptors’ response.

Line 52 - Spectral reflectance of an object does not depend on the light source, observer, or the surrounding objects. The appearance of its color does. Reflectance can however depend on the position of the observer, and it can be used together with spectral power distribution of a light source and the sensitivities of an observer to compute the color coordinates.

Line 59 – “…to represent color under light” I am not quite sure what does this mean. Either elaborate this part of the sentence, or simply omit it.

Line 62 –color models or color systems, it cannot be both. I suggest using system here, since you are mentioning Pantone later on (and Pantone is not a model).

Line 66 - “such as” should be deleted. Also, CMYK is not used to manufacture colors, but to print them.

Line 72 – Can you please clarify this sentence “These systems mainly work…”.

Line 77 – L* is lightness, not brightness. It might sound as the same thing, but it is not.

Line 82 – perceptual uniformity

Line 92 – the color complexity

Line 238 – Please add measurement unit.

7. PLOS authors have the option to publish the peer review history of their article (what does this mean?). If published, this will include your full peer review and any attached files.

Reviewer #2: **Yes: **Dr. - Ing. Vinh Quang Trinh, TU Darmstadt, Germany

Reviewer #3: No

---

## [Editor Report · Acceptance letter]

30 Apr 2021

PONE-D-20-32597R1 

Color-complexity enabled exhaustive color-dots identification and spatial patterns testing in images  

Dear Dr. Hsieh:

I'm pleased to inform you that your manuscript has been deemed suitable for publication in PLOS ONE. Congratulations! Your manuscript is now with our production department. 

Kind regards, 

on behalf of

Dr. Gulistan Raja 

Academic Editor

PLOS ONE